# A Word of Caution—Potential Limitations of Pulmonary Artery Pressure Monitoring in Detecting Congestion Caused by Right-Sided Heart Failure

**DOI:** 10.3390/biomedicines13061469

**Published:** 2025-06-14

**Authors:** Ester Judith Herrmann, Eva Herrmann, Khodr Tello, Kathleen Mantzsch, Meaza Tekeste, Stephan Fichtlscherer, Christian W. Hamm, Birgit Assmus

**Affiliations:** 1Department of Medicine I, Cardiology and Angiology, University Hospital Giessen and Marburg, 35392 Giessen, Germany; ester.herrmann@innere.med.uni-giessen.de (E.J.H.); christian.hamm@innere.med.uni-giessen.de (C.W.H.); 2German Center for Cardiovascular Research, DZHK, Partner Site Frankfurt Rhine-Main, 61231 Bad Nauheim, Germany; herrmann@med.uni-frankfurt.de; 3Institute of Biostatistics and Mathematical Modeling, Goethe University Frankfurt, 60590 Frankfurt am Main, Germany; 4Department of Medicine II, Internal Medicine, Pneumology, University Hospital Giessen and Marburg, 35392 Giessen, Germany; khodr.tello@innere.med.uni-giessen.de; 5Department of Medicine III, Cardiology, University Hospital Frankfurt am Main, 60590 Frankfurt am Main, Germany; kmantzsch@yahoo.de (K.M.); tekeste@med.uni-frankfurt.de (M.T.); fichtlscherer@em.uni-frankfurt.de (S.F.)

**Keywords:** chronic heart failure, remote monitoring, pulmonary artery pressure, right-sided decompensation

## Abstract

**Background/Objectives:** Patients with New York Heart Association (NYHA) class III heart failure (HF) suffer from frequent hospitalizations. Non-invasive pulmonary artery pressure (PAP) sensor-guided HF care has been shown to reduce hospitalizations. However, it is unknown whether the PAP changes prior to hospitalization differ between clinical right, left or global cardiac decompensation. **Methods:** Sensor-derived PAP data and HF hospitalization records from 41 patients with NYHA class III HF were classified retrospectively into predominantly left, right or global decompensation. Linear mixed-effect regression models were used for statistical evaluations of the PAP in selected hospitalizations for which admission was at least 28 days after the last admission and 14 days after the last hospital discharge and with readings in between. **Results:** During 24.4 months of follow-up, 127 hospitalizations in 38 patients were evaluated. The global cardiac decompensation (n = 13) had the highest PAP before hospitalization, followed by left-sided (n = 20) decompensation. Patients with right-sided decompensation (n = 9) had comparable PAP values before hospitalization to the cohort without any cardiac decompensation (n = 85). The diastolic PAP showed a significant increase of 0.035 mmHg/day (*p* = 0.0097) in left-sided decompensation and of 0.13 mmHg/day (*p* < 0.0001) in global cardiac decompensation, whereas no significant change in the diastolic PAP occurred prior to the right-sided decompensation. The baseline right ventricular function and right ventricle–pulmonary arterial coupling (TAPSE/PASP ratio) were impaired in patients with subsequent global cardiac decompensation. **Conclusion:** PAP telemonitoring-guided therapy can reliably detect early signs of left and global cardiac decompensation but may be limited in detecting right-sided cardiac congestion. The routine assessment of RV–PA coupling may improve the detection of global cardiac decompensation, as severe impairments could indicate impending deterioration. In contrast, monitoring the RV contractility may aid in identifying isolated right-sided congestion and imminent decompensation.

## 1. Introduction

Heart failure (HF) affects approximately 1–2% of the adult population in developed countries, and the incidence and prevalence are increasing due to the growing life expectancy [1,2]. More than 75% of acute HF hospitalizations occur in people with pre-existing chronic HF [3], so that hospitalization for acute HF is a major contributor to the economic burden of HF [4]. Moreover, every cardiac decompensation is associated with a further deterioration of myocardial function, which favors disease progression [5]. Although initial decongestion is clinically successful in the vast majority of admitted patients, 34% of acute HF patients are readmitted to hospital for the recurrence of acute HF within 3 months after discharge [6]. Consequently, there is a need for improved strategies for the early detection of HF decompensation. The monitoring of blood pressure, heart rate and weight changes as well as intrathoracic impedance as single or combined measures of congestion status were shown to have limited effectiveness in reducing hospitalization rates for HF [7,8]. However, a structured remote patient management intervention, when used in a well-defined HF population, was shown to reduce the number of days lost due to unplanned cardiovascular hospital admissions as well as all-cause mortality [9]. Likewise, results from trials with implantable hemodynamic pulmonary artery pressure (PAP) monitoring demonstrated that HF management supported by the monitoring of outpatient PAP values, as provided by the CardioMEMS^TM^ system, was associated with a significant and profound reduction in the HF hospitalization rate in New York Heart Association (NYHA) functional class III patients in the US [10,11]. More recently, the GUIDE-HF hemodynamic monitoring trial provided mixed results with a non-significant reduction in the primary endpoint in a broader patient cohort from NYHA class II-IV, which was most likely due to a significant impact on event rates by the SARS CoV-2 pandemic. In addition, the MEMS-HF trial demonstrated a significant reduction in PAP irrespective of an additional underlying precapillary component to postcapillary pulmonary hypertension (PH) [12]. More recently, the MONITOR-HF trial showed that hemodynamic monitoring substantially improved quality of life and reduced HF hospitalizations in patients with moderate-to-severe HF treated according to contemporary guidelines, which was the first large-scale randomized European study [13]. Thus, the potential of monitoring PAP to reduce HF hospitalizations is well recognized and has led to the implementation of hemodynamic telemedical care in numerous centers.

During the PAP-supported HF care of patients at our HF center, it was noticed that some patients were admitted to hospital with cardiac decompensation although there was no meaningful increase in PAP readings prior to the cardiac decompensation.

Therefore, the present case study was designed to investigate these episodes with no PAP increase prior to hospitalization for decompensation with respect to the hypothesis that the right ventricular (RV) contractility may be greater and the RV systolic function and right ventricular–pulmonary artery (RV–PA) coupling may be less impaired in these patients, compared to patients with hospitalizations with a prior increase in PAP. In addition, none of the previously published hemodynamic monitoring trials in HF have thus far provided any data with respect to RV function, contractility or RV–PA coupling.

## 2. Materials and Methods

Patients with chronic HF in NYHA functional class III and a cardiac decompensation event within the last 12 months were offered implantation of the PAP sensor (CardioMEMS^TM^, Abbott, Sylmar, CA, USA) and participation in a single-center telemonitoring registry. A total of 43 patients underwent implantation of the PAP sensor between 2015 and 2019, received PAP-guided HF management and were repeatedly trained in HF self-care by a European Society of Cardiology (ESC)-certified HF nurse. For the present analyses, one patient with an LVAD implantation and one patient with newly diagnosed constrictive pericarditis and surgery were excluded, leaving a total of 41 patients with regular data transmission of PAP values at least 5 times per week. Hospitalization was defined as an in-hospital stay ≥24 h (Figure 1).

The hospitalizations were retrospectively classified by a HF specialist based on available hospital records with clinical, imaging and laboratory criteria in order to discriminate cardiac decompensation from other reasons for hospitalization and to decide about elective and urgent cause for hospital admission. The hospitalizations for cardiac decompensation were further classified according to clinical signs and symptoms into predominately left-sided (pulmonary congestion on chest X-ray or presence of B-lines, but no peripheral congestion with edema or ascites), right-sided (peripheral edema and/or ascites, no pulmonary congestion on chest X-ray and no B-lines) or globally decompensated (evidence of pulmonary congestion on chest X-ray or B-lines and peripheral congestion with edema and/or ascites).

The outpatient routine follow-up in our Heart Failure Center took place at the defined periods of 3, 6, 12 and 24 months after PAP sensor implantation, but additional outpatient visits were possible.

RV–PA coupling was non-invasively assessed by echocardiography using the tricuspid annular plane systolic excursion/systolic pulmonary artery pressure (TAPSE/PASP) ratio method, as described previously [14]. In addition, the corresponding sensor-derived systolic PAP was used for ratio calculation. Associations between the different methods of non-invasive coupling assessment are shown in Appendix A.

Furthermore, mean pulse pressure (PP) has been calculated as average of the difference in systolic PAP and diastolic PAP and mean proportional PP is the proportion of mean PP to systolic arterial blood pressure [15].

All patients who were implanted with the CardioMEMS^TM^ sensor provided written informed consent for participation in the single-center registry (NCT03020043). This study was approved by the local ethics committee and complied with the principles laid out in the Declaration of Helsinki.

### Statistical Analysis

This case study is a descriptive comparison of four different types of hospitalizations (without cardiac decompensation event, decompensation with predominantly left-sided, right-sided and global cardiac decompensation). Baseline characteristics were compared with the non-parametric Mann–Whitney U-test. The association between TAPSE/PASP ratio (echocardiogram) and TAPSE/PASP ratio (systolic PAP derived from CardioMEMS^TM^ pressure sensor, Abbott, Sylmar, CA, USA) was analyzed by linear mixed-effect regression (Appendix A).

Linear mixed-effect models were applied for statistical evaluations of PAP and variation in PAP within the individual patients and for differentiating between the type of hospitalization in selected hospitalizations for admissions at least 28 days after the last admission, 14 days after the last hospital discharge; this left 127 hospitalizations for further analysis with respect to PAP changes (Figure 1; corresponding monitoring times prior to hospitalization are shown in Appendix A). For illustration of the time course of PAP readings, non-parametric smoothing methods were applied (kernel smoothing with the R package lokern). Thus, for the group comparisons of time-dependent, hospitalization-based variables, intra-patient dependence was considered with the mixed-effect modeling.

Statistical significance was assumed if *p* < 0.05, and all reported *p*-values are two-sided. Statistical analysis was carried out with the software packages SPSS (Version 23.0, SPSS Inc., IBM Corp., Armonk, NY, USA) and R (Version 3.3.1, R Foundation for Statistical Computing, Vienna, Austria).

## 3. Results

### 3.1. Baseline Characteristics

The hemodynamic patient cohort comprised a total of 38 patients with NYHA class III chronic HF. The mean patient age was 74 ± 2 years, 84.2% were male and 78.9% suffered from ischemic cardiomyopathy. HF with reduced ejection fraction (HFrEF) was present in 92.1% of patients. Twenty-six of thirty-eight patients (68.4%) had PH (PAPmean >20 mmHg), and of those, twenty patients presented with a combined pre- and postcapillary PH phenotype with a PVR > 2 WU, according to the ESC 2022 Guidelines [16].

During an average follow-up (with at least five PAP measurements per week) of 24.4 ± 1.3 months (range from 2 to 49 months), a total of 178 hospitalizations occurred in 38 patients (62 hospitalizations for cardiac decompensation, 116 hospitalizations for other reasons; Figure 1). Three patients in our cohort did not have any hospitalizations, thus leaving thirty-eight patients with 178 hospitalizations for further assessment. Due to the focus on the PA pressure course prior to hospitalization, we further required at least 14 days of outpatient PA pressure measurements prior to hospitalization, resulting in 127 hospitalizations in 38 patients for further evaluation (Appendix A).

These hospitalizations included 42 hospitalizations for HF in 16 patients, which were further sub-classified as predominantly left-sided (n = 20), right-sided (n = 9) or global (n = 13) cardiac decompensation events. A total of 85 hospitalizations occurred without cardiac decompensation (Figure 1).

Eleven patients died during the follow-up period, with eight patients experiencing cardiac death, but no patient died within 30 days after hospital admission (Figure 2 and Appendix A).

The analysis of the association of a later cardiac decompensation event with patient baseline characteristics revealed that these events are associated with higher creatinine levels and a more impaired RV–PA coupling, as assessed non-invasively by the TAPSE/PASP ratio at the time of the PAP sensor implantation (Table 1). With respect to the hemodynamics of HF, the cardiac index of the cohort with cardiac decompensation was comparable to that of the group without cardiac decompensation, although the PAP was elevated in the former group. The pulmonary arterial wedge pressure was similarly elevated in the two groups, and the pulmonary vascular resistance was pathologically elevated in 52% of the hospitalizations.

The length of hospital stays was longer in all types of cardiac decompensation than in the hospitalizations without cardiac decompensation (left-sided: 8.0 ± 2.7 days; *p* = 0.046, right-sided: 5.3 ± 1.6 days; *p* = 0.731, global: 14.8 ± 3.14 days; *p* = 0.083, vs. without cardiac decompensation event: 4.5 ± 0.6 days). Hospitalizations for the global cardiac decompensation had the longest duration of hospitalization (Table 2).

The comparison of baseline variables between the different clinical types of cardiac decompensation revealed that clinical and functional parameters were worse in cases with global cardiac decompensation events. For example, patients with a right-sided or global cardiac decompensation during follow-up tended to be older, had higher levels of the N-terminal fragment of the pro-brain natriuretic peptide (NT-proBNP) and worse kidney function than patients with left-sided cardiac decompensation events (Table 1). The RV–PA coupling at baseline, measured as the TAPSE/PASP ratio, was significantly impaired in global versus right-sided cardiac decompensation (TAPSE/PASP ratio 0.40 ± 0.05 in right-sided decompensation versus 0.26 ± 0.04 in global decompensation events, *p* = 0.038). The RV–PA coupling in patients with global cardiac decompensation was also numerically lower in comparison to left-sided cardiac decompensation, although the difference was not significant (TAPSE/PASP ratio 0.36 ± 0.05 in left-sided decompensation versus 0.26 ± 0.04 in global decompensation events, *p* = 0.211). When the TAPSE/PASP ratio was calculated using both the echocardiographic- and sensor-derived systolic PAP, the latter value was associated with numerically more cases due to limitations in echocardiographic assessments. However, the two calculated ratios showed a highly significant positive association with each other (at baseline: R^2^ = 0.33, *p* = 0.001, and at last visit prior to hospitalization: R^2^ = 0.34, *p* < 0.001; Appendix A).

### 3.2. Association of Follow-Up Characteristics with Subsequent Decompensation Events

The impact of more recent clinical, laboratory and echocardiographic data, which were obtained at the follow-up visits, was assessed with respect to the impending cardiac decompensation event. These analyses showed that the pattern of differences obtained at baseline persisted during the follow–up. For example, NT-proBNP levels remained significantly lower in patients with left-sided cardiac decompensation than in those with right-sided or global cardiac decompensation (left-sided 3.3 ± 0.13 pg/mL versus global 3.7 ± 0.25 pg/mL, *p* = 0.012, left-sided versus right-sided 3.6 ± 0.19 pg/mL, *p* = 0.024), and the kidney function was worse in patients with global cardiac decompensation (Table 2). Likewise, in comparison with the baseline, there was no change in the RV–PA coupling pattern or RV function in the left-sided or global cardiac decompensation groups and just a slight deterioration of the RV function prior to right-sided decompensation events. However, a limitation of this analysis is that there are different time frames between the last observation obtained at the follow-up visit and the impending cardiac decompensation event.

### 3.3. PAP Course Prior to Different Types of Cardiac Decompensation

Based on the course of the outpatient PAP values in the 30 days prior to a non-cardiac hospitalization event, there was a significant decrease in the systolic PAP and mean PAP of (mean ± standard error) 0.037 ± 0.01 mmHg/day (*p* = 0.0013) and 0.022 ± 0.008 mmHg/day (*p* = 0.0074). In contrast, there was a significant increase in the diastolic PAP of 0.035 ± 0.01 mmHg/day (*p* = 0.0104) in patients with predominantly left-sided decompensation and a significant increase in the systolic PAP, diastolic PAP and mean PAP of 0.14 ± 0.03 mmHg/day (*p* < 0.0001), 0.12 ± 0.02 mmHg/day (*p* < 0.0001) and 0.12 ± 0.02 mmHg/day (*p* < 0.0001), respectively, in patients with global decompensation. However, no significant changes in PAP values occurred prior to predominantly right-sided decompensation events (0.003 ± 0.02 (*p* = 0.90)). Notably, the models account for a slightly but significantly reduced standard deviation in patients with right-sided decompensation by 25% (*p* < 0.0001), 20% (*p* = 0.0005) and 24% (*p* < 0.0001) in the systolic PAP, diastolic PAP and mean PAP, respectively. The graphical representations (Figure 3top) show the smoothed PAP curves for the time course.

The mean PP in left-sided and global cardiac decompensation was higher compared to no cardiac decompensation (*p* = 0.006 and *p* < 0.0001 for left and global, respectively), while there were no significant differences for the mean proportional PP. Furthermore, there was a significant decrease in the mean PP of 0.03 ± 0.01 mmHg/day (*p* = 0.0011) but not in the mean proportional PP in patients without cardiac decompensation. Again, models account for a significantly reduced standard deviation in patients with right-sided decompensation, by 19% (*p* = 0.0011) and 71% (*p* < 0.0001) for the mean PP and mean proportional PP, respectively. In all other cases, there was no significant trend.

Exemplarily, the course of PAP values 30 days prior to hospitalization is shown for the four different types of hospitalization as a single case study in Figure 3bottom.

## 4. Discussion

The present retrospective analysis is, to our knowledge, the first report to investigate the different clinical types of cardiac decompensation in a patient cohort of NYHA class III HF managed by PAP-supported HF care. Our case study confirms that PAP telemonitoring can reliably detect early signs of left and global cardiac decompensation but suggests that it may be less effective in detecting predominately right-sided congestion followed by decompensation events because there may be a lack of a PAP increase prior to cardiac decompensation in these patients.

Notably, we detected a significantly reduced standard deviation as an expression of a reduced variability of the PA pressure, mean PP and mean proportional PP only in patients with right-sided decompensation events but not in patients with left-sided or global cardiac decompensation. This has not yet been described previously and might represent a marker of a reduced RV stroke volume and contractility, aligning with the interpretation of proportional PP as a surrogate marker for RV function proposed by Mazimba et al. [16].

In patients with HF, the presence of secondary PH is associated with a reduced exercise capacity and worse prognosis [17,18,19,20]. This might be explained if one considers that PH reflects more advanced HF, leading to a more upstream transmission of the increased pulmonary artery wedge pressure to PAP, which in turn favors the development of afterload-induced RV failure. As we know that RV afterload consists of a static (PVR) and a combined static and dynamic pulsatile (pulmonary arterial capacitance as a ratio of the stroke volume over pulmonary PP) component [21,22], therefore, it could be anticipated that there might be differences in the clinical presentation between the types of cardiac decompensation depending on which component of the RV afterload is predominantly affected.

However, PA pressure monitoring can successfully be used in patients with type 1 PH, as demonstrated by Benza et al. [23]. In this group with NYHA functional class III-IV symptoms, the cardiac index was not markedly reduced, and separate data on the RV function or RV–PA coupling were not reported. Nevertheless, since RV function is exquisitely sensitive to afterload, afterload-reducing treatments and management, like the present PAP-guided HF treatment, could provide substantial benefits in different types of PH. Notably, a subgroup analysis of the MEMS-HF registry demonstrated that patients without PH, but also those with isolated PH and even patients with combined pre-and postcapillary PH, derived similar benefits from PAP-guided HF care [12].

Interestingly, a recent study by Nies et al of 114 patients with paradoxical low-flow, low-gradient aortic stenosis demonstrated that right ventricular dysfunction (RVD) and impaired RV–PA coupling were present in 50% of cases and independently associated with increased post-transcatheter aortic valve replacement (TAVI) mortality [24]. These findings support our observations by underscoring the prognostic relevance of the RV function and RV–PA coupling, even in patient populations where the left ventricular function appears preserved. Together, they highlight the need for comprehensive biventricular evaluations in heart failure management and reinforce the value of RV dysfunction and RV–PA coupling as a prognostic marker across different cardiovascular pathologies.

The prevention of RV functional deterioration and RV–PA uncoupling might be an additional mechanism by which PAP-guided HF management may improve outcomes in heart failure patients. It might therefore be interesting to serially assess the RV function and contractility in future trials with hemodynamically guided HF treatments. In our small case study, the routinely performed echocardiographic assessments during follow-up showed no changes in the RV function and RV–PA coupling in predominantly right-sided decompensation. But, we observed a significantly reduced day-to-day variation in the PAP and of the mean PP and mean proportional PP prior only to the right-sided decompensation, which may be an indirect sign for a reduced right ventricular contractility, distensibility and pulmonary arterial capacitance (PAC) [15]. Thus, predominantly right-sided decompensation events might rather reflect a decrease in RV contractility than an impairment of the systolic RV function and RV–PA uncoupling. Whether additional assessments of RV contractility can help to prevent hospitalization in cases of right-sided decompensation without a corresponding rise in PAP should be investigated in future studies.

However, a reduced TAPSE/PASP ratio and TAPSE, even at the baseline and prior decompensation, was identified as a possible surrogate parameter for pending global cardiac decompensation. Importantly, the TAPSE/PASP ratio seemed to provide more impact if calculated with the hemodynamically obtained PAP, which may be related to better accuracy. In line with our findings, Boehm et al. showed in an experimental, sham controlled study, that isolated RV pressure overload induced by pulmonary artery banding induces RV–PA uncoupling and reduces the systolic RV function and even RV hypercontractility [25]. Therefore, we hypothesize that a predominantly right-sided decompensation might rather be a result of a reduced RV contractility, thus without an increase in PAP, and a predominantly global cardiac decompensation may rather possibly be a result of a longer-term increase in PAP with a subsequent RV–PA uncoupling and deterioration of RV function. However, this indeed needs further investigations in larger trials.

In general, the development of RV dysfunction and failure is regarded as a turning point in the natural course of heart failure and is associated with a worse outcome [26]. Most of the previous work related to adaptive and maladaptive RV remodeling and failure has been conducted in the setting of pulmonary arterial hypertension (Group 1 PH), whereas most patients with PH suffer from left-sided heart disease (Group 2 PH). The presence of early RV dysfunction is challenging to detect in the routine clinical settings. While Gruenig et al. [27] proposed an exercise-induced increase in systolic PAP as a surrogate marker of the RV contractile reserve, a more recent analysis proposed to use an exercise RV ejection fraction to non-invasively detect occult RV dysfunction [28]. In a further analysis of the CHAMPION trial, a low proportional PP identified patients with low forward flow (stroke volume) and a low PA PP (RV contractility and distensibility) [15]. However, the most sensitive and precise way to measure the RV contractile reserve is RV–PA coupling [29,30,31]. RV–PA coupling is defined by RV E_ES_/Ea, or the ratio of the RV end-systolic elastance (E_ES_) to the PA effective arterial elastance (Ea). RV E_ES_ is a load-independent method to measure intrinsic RV contractility, whereas E_ES_/Ea measures the right-sided ventriculo-arterial coupling. The optimal Ees/Ea ratio is between 1.5 and 2.0. RV–PA coupling in PH has considerable reserve, and the Ees/Ea threshold at which uncoupling occurs is estimated to be approximately 0.7 [31]. The gold-standard method for measuring RV–PA coupling is the conductance catheter-based multi-beat assessment. However, Tello et al. showed that the non-invasive, echocardiographic TAPSE/PASP ratio could be used for the assessment of RV–PA coupling in patients with severe pulmonary arterial hypertension [14]. More recently, this ratio was incorporated into the updated ESC guidelines for PH, specifically into the echocardiographic risk stratification of patients with Group 1 PH [16]. Although the groundbreaking work was performed in Group 1 PH patients, numerous trials have shown that the TAPSE/PASP ratio also provides important information about the right ventricle, RV remodeling and prognosis in left-sided heart disease. For example, in a retrospective cohort study of 228 patients after a MitraClip^®^ application for severe mitral regurgitation, a TAPSE/PASP ratio below the median value of 0.35 mm/mmHg was associated with post-intervention adverse events like HF hospitalization and all-cause mortality [32]. In a larger cohort of 1.149 cardiology patients from Leiden University with moderate to severe secondary tricuspid regurgitation, the median TAPSE/PASP ratio was 0.35 mm/mmHg at the time of the diagnosis of significant secondary tricuspid regurgitation [33]. During a 5-year follow-up, RV–PA uncoupling, defined as a TAPSE/PASP ratio < 0.31 mm/mmHg, was the only echocardiographic parameter independently associated with all-cause mortality. Interestingly, most patients in our cohort who experienced cardiac decompensation also had a TAPSE/PASP ratio < 0.35 mm/mmHg, indicating an increased risk and advanced HF, especially the group of patients with global cardiac decompensation. In contrast, hospitalizations without cardiac decompensation had normal RV–PA coupling (0.50 ± 0.03 mm/mmHg). Thus, the routine measurement of RV–PA coupling could be used for pending global cardiac decompensation.

Moreover, in contrast to other studies, we calculated the TAPSE/PASP ratio by two methods, using the traditional TAPSE as well as either the routine echocardiography-derived PASP value or the directly measured sensor-derived systolic PAP. Thus, the PAP could therefore be obtained immediately prior to the cardiac decompensation event, for a “de novo” TAPSE/PASP ratio during follow-up. However, although a significant positive association between the traditional pure echocardiographic and the hemodynamic-supported TAPSE/PASP ratio persisted, there were no substantial changes between the first baseline-derived TAPSE/PASP ratio and the follow-up TAPSE/PASP ratio, which may be related to the fact that the TAPSE values were collected at pre-defined time points with variable time intervals to the pending cardiac decompensation event.

Finally, the ultimate goal of measuring congestion is obviously not yet achieved by using a hemodynamic sensor for measuring pressure as a surrogate of resistance and volume. Nonetheless, clinically established alternatives are rare, and the recently presented CAVA-ADHF-DZHK10 trial (ESC 2023) could not demonstrate that the ultrasound-guided imaging of the inferior Vena cava to guide decongestion in ADHF patients was superior to the standard treatment [34]. Thus, the optimal assessment of congestion remains to be determined. Experimentally, it has been shown that the filling pressure cannot always be used as a specific parameter for the evaluation of the intravascular volume [35], which might be of specific interest in the context of right-sided HF. A recent report demonstrated a lack of correlation between different types of congestion markers in acute decompensated HF [36], which underlines the difficulties of the early detection of right heart decompensation.

## 5. Limitations

The limitations of the case study, in addition to it being a retrospective analysis, are the small patient cohort and the limited number of hospitalizations. The classification of the type of HF decompensation event was performed by a single HF-experienced clinician based on the medical report of clinical, imaging and laboratory data. Most importantly, patients were treated for increasing PAP values to avoid cardiac decompensation, and we cannot exclude that other potentially pending right-sided or global cardiac decompensations had elevations in PAP that triggered treatment and, thereby, that hospitalization could be prevented. Prior to the predominantly left-sided decompensation, there were 15 down-titrations of diuretics and 3 of the guideline-directed medical therapy (GDMT), compared to 27 up titrations of diuretics and 3 of GDMT. Prior to the global cardiac decompensations, we identified eight down-titrations of diuretics and two of GDMT, compared to sixteen up-titrations of diuretics and one of GDMT. In the cases of predominantly right-sided decompensation, there have been just two down-titrations of diuretics and one of GDMT, compared to seven up-titrations of diuretics and zero of GDMT (Appendix A). Thus, the analyzed cardiac decompensation events occurred despite the telemedical advice for ambulatory medical treatment changes, which may be an indicator for advanced HF requiring even more intense treatment, such as the intravenous administration of diuretics.

In addition, the follow-up measurements were collected at the defined dates after the PAP sensor implantation; thus, the last available date of assessments prior to hospitalization varied.

## 6. Conclusions

Our case study shows that outpatient daily PAP-guided HF therapy can reliably detect early signs of impending left and global ventricular decompensation but may have limitations in detecting predominately right-sided decompensation.

The variability of the PA pressure, PP and proportional PP was profoundly reduced in these patients; however, whether this can be used as a predictive additive tool for the identification of the impeding deterioration of RV contractility and decompensation was beyond the scope of this analysis. Future studies with a prespecified RV contractility analysis, echocardiographic parameters (e.g., TAPSE; TAPSE/PASP ratio, RV-FAC) and machine learning will provide more insight.

Furthermore, patients could profit from an additional routine measurement of RV–PA coupling, because a severely impaired RV–PA coupling might be a hint of pending global cardiac decompensation. PAP sensor trials, like the ongoing PASSPORT trial in Germany, routinely collect data on RV function; thus, more detailed insights into the value of the PA pulse pressure and RV–PA coupling for hemodynamic HF care, but also for standard nurse-based care, will be available upon the trial’s completion.

## Figures and Tables

**Figure 1 biomedicines-13-01469-f001:**
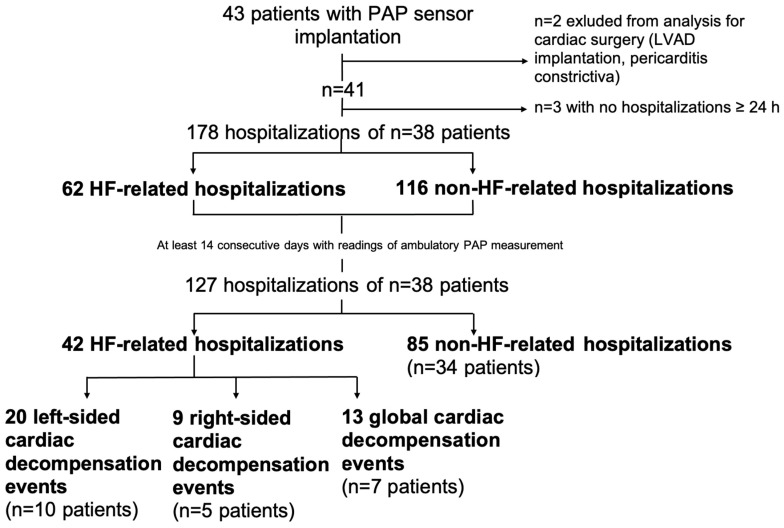
Flowchart of hospitalization analysis.

**Figure 2 biomedicines-13-01469-f002:**
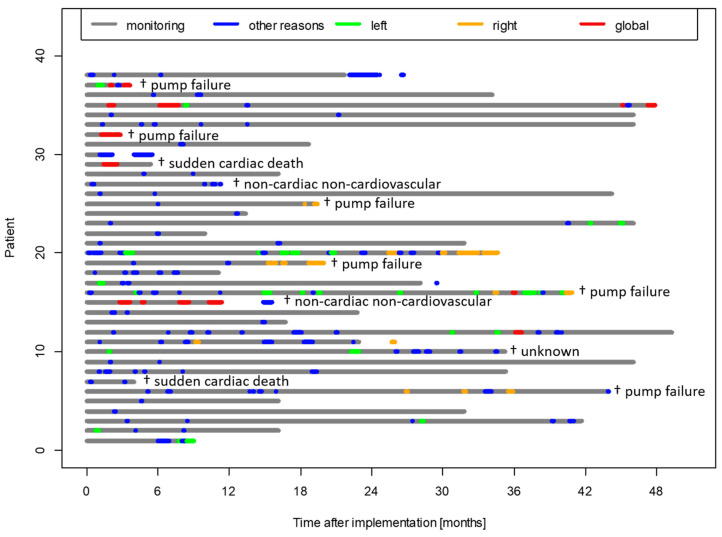
Graphical display of monitoring of patients with decompensation events (n = 38), including cause of death (n = 11).

**Figure 3 biomedicines-13-01469-f003:**
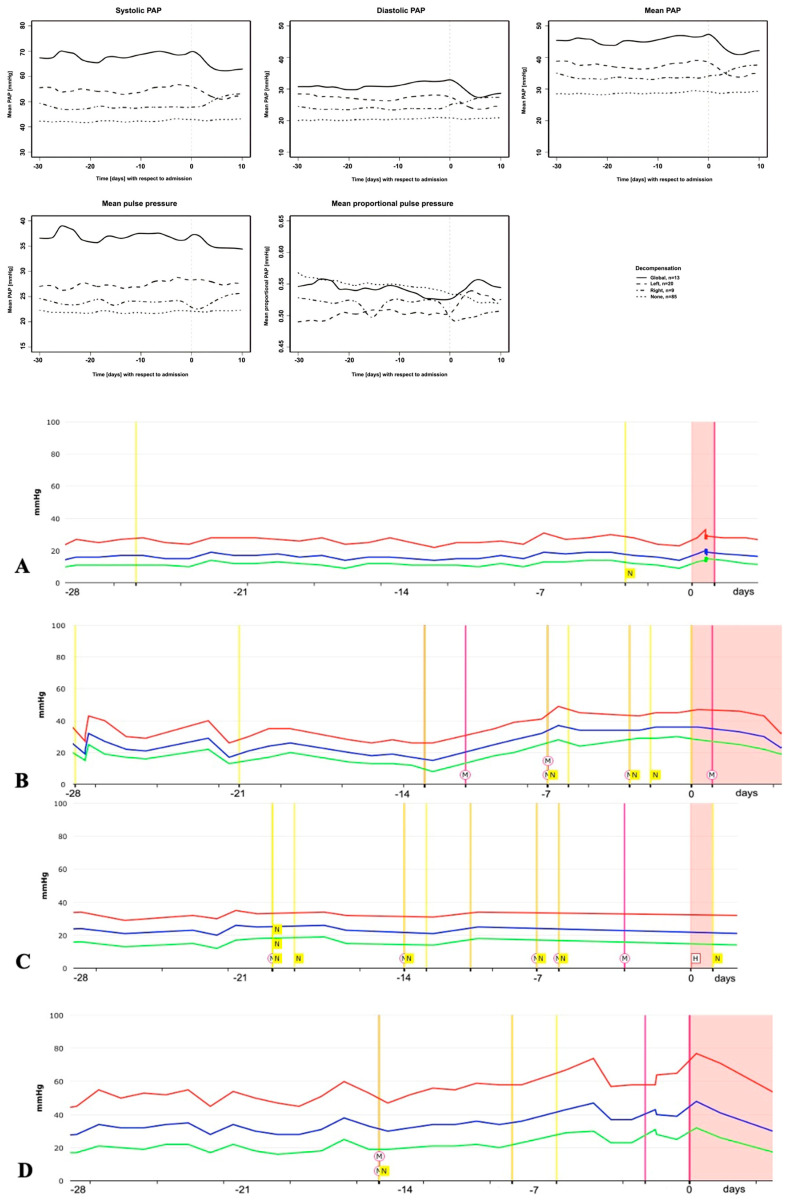
(**Top**) the time course of pulmonary artery pressure (PAP) values in the 30 days prior to hospitalization. The graphical representation shows smoothed mean curves for the time course. (**Bottom**) examples of patients’ PAP course (raw data) prior to hospitalization. (**A**): hospitalization for non-cardiac decompensation; (**B**): predominantly left-sided decompensation; (**C**): predominantly right-sided decompensation; and (**D**): global cardiac decompensation. Day 0 = day of hospitalization. Red line, systolic PAP. Blue line, mean PAP. Green line, diastolic PAP. Pink bar with circled “M”, medication adjustment. Yellow bar with “N”, denotes telephone contact with patient.

**Table 1 biomedicines-13-01469-t001:** Patient baseline characteristics analyzed per hospitalization at the time of the sensor implantation.

Variables	Hospitalizations Without Cardiac Decompensation (n = 85;%)	Hospitalizations with Cardiac Decompensation (n = 42;%)	*p*-Value	Left-Sided Decompensation (n = 20;%)	Right-Sided Decompensation (n = 9;%)	Global Cardiac Decompensation (n = 13;%)	*p*-Values ^1,2,3^ *p*-Values ^a,b,c^
**Clinical characteristics**							
Age, years	74 ± 1	79 ± 1	0.098	76 ± 2	80 ± 1	81 ± 1	0.614/0.283/0.738 0.381/0.283/0.940
Etiology of CHF							
HFpEF	2; 8%	2; 6%	0.546	1; 9%	0; 0%	1; 20%	0.456/0.989/0.589
HFrEF	83; 92%	40; 94%		19; 91%	9; 100%	12; 80%	0.992/0.739/0.984
NYHA class							
IIIA	83; 94%	36; 88%	0.079	19; 87%	8; 80%	9; 75%	0.509/0.301/0.670
IIIB	2; 6%	6; 12%		1; 13%	1; 20%	4; 25%	0.595/0.174/0.059
Body Mass Index (kg/m^2^)	26 ± 0.3	27 ± 0.4		27 ± 0.7	26 ± 0.8	27 ± 0.7	
Systolic blood pressure, mmHg	114 ± 2	112 ± 2	0.149	111 ± 4	110 ± 4	109 ± 3	0.622/<**0.001**/0.729 0.339/0.252/0.903
**Laboratory parameters**							
Creatinine, mg/dl	1.44 ± 0.04	1.79 ± 0.10	**0.021**	1.75 ± 0.12	1.40 ± 0.09	2.27 ± 0.21	0.204/0.698/0.086 0.337/**0.030/**0.219
Log-NT-proBNP, pg/mL	3.2 ± 0.07	3.5 ± 0.08	0.079	3.2 ± 0.07	3.6 ± 0.13	3.5 ± 0.19	0.403/**0.046**/0.551 **0.006**/**0.007**/<**0.001**
**Medication at baseline**							
ACEI/ATRB	70; 78%	41; 92%	0.085	20; 100%	9; 100%	12; 86%	0.995**/**0.997/0.966 1.000/0.983/0.987
ARNI	15; 20%	1; 2%	0.085	0; 0%	0; 0%	1; 14%	0.987**/**0.98/0.965 1.000/0.997/0.999
ACEI/ARB/ARNI	85; 100%	42; 100%	1.000	20; 100%	9; 100%	13; 100%	1.000/1.000/1.000 1.000/1.000/1.000
Beta-blockers	83; 97%	40; 91%	0.470	18; 92%	9; 100%	13; 100%	0.200/0.998/0.976 0.982/1.000/1.000
MRA	57; 81%	30; 69%	0.468	15; 70%	9; 100%	6; 40%	0.922/0.969**/**0.511 0.991/0.218**/**0.911
Loop diuretics	77; 92%	42; 100%	0.996	20; 100%	9; 100%	13; 100%	0.972/0.997/0.989 1.000/1.000/1.00
Torasemide, dose equivalent in mg	22 ± 2	26 ± 2	0.447	26 ± 3	23 ± 4	30 ± 4	0.494/0.901/0.680 0.661/0.734/0.550
Furosemide, dose equivalent in mg	88 ± 8	104 ± 8		104 ± 12	92 ± 16	120 ± 16	
**Echocardiography**							
LVEF, %	26 ± 1	25 ± 2	0.598	28 ± 3	24 ± 3	29 ± 3	0.688/0.380/0.781 0.324/0.731/**0.079**
TAPSE, mm	18 ± 1	17 ± 1	0.070	16± 1	21 ± 2	13 ± 1	0.080/0.154/<**0.001** 0.161**/0.014/**0.157
RV fractional area change, %	39 ± 1.1	38 ± 2.4	0.131	41 ± 3.2	48 ± 6.9	36.2 ± 9.9	**0.001**/0.277/0.712 0.894/0.201/0.476
TAPSE/PASP ratio, mm/mmHg (Echo)	0.52 ± 0.04	0.42 ± 0.03	0.190	0.42 ± 0.05	0.37 ± 0.08	0.28 ± 0.09	0.334**/**0.578/0.099 0.140/0.807/0.454
TAPSE/PASP ratio, mm/mmHg (CardioMEMS^TM^)	0.50 ± 0.03	0.35 ± 0.03	**0.007**	0.36 ± 0.05	0.40 ± 0.05	0.26 ± 0.04	0.060/0.351/**0.045** 0.626/0.211**/0.038**
Association TAPSE/PASP ratio (Echo vs. CardioMEMS^TM^)	*p* = **0.0012**	*p* = **0.0014**	-	*p* = 0.423	*p* < **0.001**	*p* = 0.228	-
**Invasive hemodynamics**							
Cardiac index, l/min/m^2^	2.0 ± 0.1	2.0 ± 0.2	0.687	2.6 ± 0.3	2.2 ± 0.5	1.8 ± 0.1	0.206/0.636/0.768 0.760/0.197/0.062
PAP systolic, mmHg	44 ± 2	53 ± 2	**0.002**	53 ± 4	54 ± 4	54 ± 5	**0.030**/0.169/0.564 0.940/0.422/0.754
PAP mean, mmHg	27 ±1	34 ± 2	**0.013**	33 ± 3	34 ± 3	34 ± 3	0.099/0.262/0.667 0.879/0.495/0.804
PAP diastolic, mmHg	17 ± 1	22 ± 1	**0.031**	20 ± 2	23 ± 2	22 ± 2	0.261/0.194/0.608 0.561/0.418/0.901
PCWP mean, mmHg	17 ±1	18 ± 1	0.524	19 ± 2	16 ± 3	18 ± 2	0.870/0.396/0.958 0.396/0.897/0.483
PVR, Wood unit	3.5 ± 0.3	2.7 ± 0.2	**0.017**	3.1 ± 0.4	3.0 ± 0.7	2.4 ± 0.2	0.238/0.386/0.625 0.796/0.170/0.074
cpcPH, n, %	43; 51%	23; 55%	-	14; 70%	6; 67%	3; 23%	-

Values represent the mean ± standard error or n (%) from linear or logistic mixed-effect regression models accounting for multiple hospitalizations in single patients. First *p*-values are for the association with the type of cardiac decompensation with the reference group of hospitalizations without cardiac decompensation (^1^ *p*-value: no cardiac decompensation vs. left, ^2^ *p*-value: no cardiac decompensation vs. right and ^3^ *p*-value: no cardiac decompensation vs. global), and reported *p*-values are two-sided and not corrected for multiple comparisons. Second *p*-values are for the association with different types of cardiac decompensation (^a^ *p*-value: left vs. right, ^b^ *p*-value: left vs. global and ^c^ *p*-value: right vs. global), and *p*-values are two-sided and not corrected for multiple comparisons. The association between the TAPSE/PASP ratio (echocardiogram) and TAPSE/PASP ratio (CardioMEMS^TM^) was also analyzed with a mixed-effect regression model, and the R^2^ is the proportion of variations explained by the fixed effect factors, and *p*-values are two-sided. Baseline medication: ACEI-I, angiotensin-converting enzyme inhibitor; ARNI, angiotensin receptor blocker neprilysin inhibitor; ATRB, angiotensin receptor blocker; and MRA, mineralocorticoid antagonist. Other abbreviations: CHF, chronic heart failure; HFpEF, heart failure with preserved ejection fraction; HFrEF, heart failure with reduced ejection fraction; NYHA, New York Heart Association; NT-proBNP, N-terminal fragment of pro-brain natriuretic protein; LVEF, left ventricular ejection fraction; TAPSE, tricuspid annular plane systolic excursion; RV, right ventricular; PASP, systolic pulmonary artery pressure; PAP, pulmonary artery pressure; PCWP, pulmonary capillary wedge pressure; PVR, pulmonary vascular resistance; and cpcPH, combined post- and precapillary pulmonary hypertension.

**Table 2 biomedicines-13-01469-t002:** Clinical characteristics and echocardiographic and laboratory parameters pre-decompensation without cardiac decompensation vs. pre-decompensation per hospitalization for all cardiac decompensation events and for those classified as left-sided, right-sided and global cardiac decompensation.

Variables	Hospitalizations Without Cardiac Decompensation (n = 85;%)	Hospitalizations with Cardiac Decompensation (n = 42;%)	*p*-Value	Left-Sided Decompensation (n = 20;%)	Right-Sided Decompensation (n = 9;%)	Global Cardiac Decompensation (n = 13;%)	*p*-Values ^1,2,3^ *p*-Values ^a,b,c^ *p*-Values *^,§,$^
**Clinical characteristics**							
NYHA class							
IIA	18; 12%	15; 19%	0.0788	9; 18%	4; 40%	2; 30%	
IIB	4; 3%	0; 0%		0; 0%	0; 0%	0; 0%	0.186 ^1^/0.236 ^2^/0.208 ^3^
IIIA	61; 80%	23; 67%		10; 72%	4; 50%	9; 58%	0.559 ^a^/0.572 ^b^/0.927 ^c^
IIIB	2; 6%	4; 13%		1; 10%	1; 10%	2; 12%	
**Echocardiographic parameters**							
TAPSE, mm	19.1 ± 0.6	17.0 ± 1.1	**0.010**	18.1 ± 1.3	17.7 ± 0.7	14.2 ± 1.5	0.272 ^1^/0.357 ^2^/**<0.001** ^3^ 0.686 ^a^/**0.014** ^b^/0.235 ^c^
TAPSE/PASP ratio, mm/mmHg (Echo)	0.53 ± 0.04	0.42 ± 0.05	**0.030**	0.47 ± 0.06	0.41 ±0.04	0.28 ± 0.04	0.333 ^1^/0.137 ^2^/0.057 ^3^ 0.336 ^a^/**0.030** ^b^/0.360 ^c^
TAPSE/PASP ratio, mm/mmHg (CardioMEMS^TM^)	0.53 ± 0.04	0.37 ± 0.06	**0.014**	0.39 ± 0.08	0.37 ± 0.05	0.31 ± 0.04	0.159 ^1^/0.114 ^2^/**0.028** ^3^ 0.642 ^a^/0.098 ^b^/0.352 ^c^
Association TAPSE/PASP ratio (Echo vs. CardioMEMS^TM^)	*p* < **0.001**	*p* < **0.001**	-	*p* = 0.894	*p* = 0.999	*p* = 0.981	-
RV fractional area change, %	37 ± 1	34 ± 2	0.066	36 ± 2	35 ± 2	30 ± 2	0.491 ^1^/0.502 ^2^/0.219 ^3^ 0.620 ^a^/**0.034** ^b^/0.463 ^c^
**Laboratory parameters**							
Creatinine, mg/dl	1.61 ± 0.08; ^#^ *p* **<** 0.001	2.40 ± 0.45; ^&^ *p* **=** 0.025	**0.021**	2.51 ± 0.74	1.91 ± 0.21	3.07 ± 0.90	0.345 ^1^/ 0.241 ^2^/0.319 ^3^ 0.922 ^a^/ 0.187 ^b^/0.482 ^c^ 0.176 */0.142 **^§^**/0.231 ^$^
Rel. Creatinine, %	17% ± 3%	33% ± 16%	0.310	33% ± 26%	35% ± 13%	40% ± 33%	0.953 ^1^/0.095 ^2^/0.877 ^3^ 0.391 ^a^/0.836 ^b^/0.867 ^c^
Log-NT-proBNP, pg/ml	3.2 ± 0.08; ^#^ *p* = 0.172	3.6 ± 0.13; ^&^ *p* = 0.228	**0.015**	3.2 ± 0.13	3.6 ± 0.19	3.7 ± 0.25	0.870 ^1^/0.067 ^2^/0.310 ^3^ **0.024** ^a^/**0.012** ^b^/0.861 ^c^ 0.305 *****/0.414 **^§^**/0.478 ^$^
Δ log-NT-proBNP, pg/ml	−0.06 ± 0.04	0.12 ± 0.10	0.098	0.16 ± 0.14	0.07 ± 0.08	0.16 ± 0.21	0.318 ^1^/0.404 ^2^/**0.016** ^3^ 0.623 ^a^/0.945 ^b^/0.487 ^c^
Urgent admission, n; %	60; 78% 23; 22%	22; 67% 16; 33%	0.311	10; 70% 10; 30%	5; 60% 4; 40%	7; 67% 2; 33%	0.291 ^1^/0.498 ^2^/0.931 ^3^ 0.860 ^a^/0.283 ^b^/0.692 ^c^
Length of hospitalization, days	4.5 ± 0.6	11.0 ± 1.8	**0.003**	8.0 ± 2.7	5.3 ± 1.6	14.8 ± 3.14	**0.046** ^1^/ 0.731 ^2^/0.083 ^3^ 0.437 ^a^/0.073 ^b^/0.292 ^c^

Values represent the mean ± standard error or n (%). The last available observation before hospitalization is reported. First *p*-values are for the association of the type of cardiac decompensation with the reference group of hospitalizations without cardiac decompensation (^1^ *p*-value: no cardiac decompensation vs. left, ^2^ *p*-value: no cardiac decompensation vs. right, ^3^ *p*-value: no cardiac decompensation vs. global) using a mixed-effect logistic regression analysis; reported *p*-values are two-sided and not corrected for multiple testing. Second *p*-values are for the association of the types of cardiac decompensation (^a^ *p*-value: left vs. right, ^b^ *p*-value: right vs. global, ^c^ *p*-value: left vs. global) using a mixed-effect logistic regression analysis; *p*-values are two-sided and are not corrected for multiple testing. Third *p*-values are for the comparison from the baseline versus pre-hospitalization within the different types of hospitalization (baseline vs. pre-hospitalization: ^#^ *p*-value for hospitalizations without cardiac decompensation events; ^&^ *p*-value for all cardiac decompensation events; * *p*-value for left-sided decompensation events; ^§^ *p*-value for right-sided decompensation events; and ^$^ *p*-value for global cardiac decompensation events) using linear logistic regression for log ratios. The last observation and relative change from baseline and to the last observation in laboratory parameters prior to hospitalization. The association between the TAPSE/PASP ratio (echocardiogram) and TAPSE/PASP ratio (CardioMEMS^TM^) was also analyzed with a mixed-effect regression model, and *p*-values are two-sided. Further characteristics of the hospitalization and the outcome per hospitalization. Mean and standard errors or proportions from a corresponding linear or logistic random effect regression estimate. For abbreviations see Table 1.

## Data Availability

The data are combined together and analyzed within the single-center registry (NCT03020043) and can be obtained by a written request.

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
