# Peer review of "A Word of Caution—Potential Limitations of Pulmonary Artery Pressure Monitoring in Detecting Congestion Caused by Right-Sided Heart Failure"

_biomedicines, 2025, doi:10.3390/biomedicines13061469_

Round 1
Reviewer 1 Report
Comments and Suggestions for Authors
Dear authors of the paper Biomedicines-3635712, I offer constructive recommendations regarding your work.
Line 44: Please describe the abbreviation NYHA.
Line 44: Please use the abbreviations you described.
Lines 46-47: Please use the abbreviations you described.
Line 52: Please use the abbreviations you described.
Line 56: Please use the abbreviations you described.
Line 58: Please use the abbreviations you described.
Line 61: Please use the abbreviations you described.
Line 82: Please describe the abbreviation RV-PA.
Line 89: Please use the abbreviations you described.
Line 93: Please use the abbreviations you described. described
Lines 93-95: Please use the abbreviations you described.
Line 95: Please use the abbreviations you described.
Line 102: Please use the abbreviations you described.
Line 115: You had already used the abbreviation RV-PA, but you did not describe it.
Line 156: Please use the abbreviations you described.
Line 188: Please use the abbreviations you described.
Lines 137-238: Please describe the abbreviation.
Lines 237-238: Please describe the abbreviation NT-proBMP.
Line 239: Please use the abbreviations you described.
Lines 289-292-296-317 Please use the abbreviations you described.
Line 310 Please use the abbreviations you described.
Line 312 Please correctly discuss the bibliographic citation (17), do not just include a number.
Line 322 Please use the abbreviations you described.
Line 322 You had already described the abbreviation PH.
Line 350 Please use the abbreviations you described.
Line 352 Please correctly discuss the bibliographic citation (17), do not just include a number.
Line 379 Please use the abbreviations you described.
Author Response
Reviewer #1:
Reviewer Comment: Line 44: Please describe the abbreviation NYHA.
Response: This comment might be accidentally mixed up with the next comment. In line 44, we mention HF as “heart failure” which is now explained on its first occurrence, and consequently substituted by the abbreviation “HF” (line 49 in the revised manuscript).
The abbreviation NYHA (New York Heart Association) has been defined in the manuscript on page 1, line 18.
Reviewer Comment: Line 44: Please use the abbreviations you described.
Response: We have ensured that all described abbreviations, including NYHA, are consistently used throughout the manuscript from this point onward.
Reviewer Comments (Lines 46–47, 52, 56, 58, 61, 89, 93, 93–95, 95, 102, 156, 188, 310, 322): Please use the abbreviations you described.
Response: Thank you. We confirm that the abbreviation HF (heart failure) is used consistently throughout the text (lines 50, 52-53, 57, 59, 62-63, 66, 68, 76-77, 79, 82, 90, 113, 117-118, 125, 185, 204, 221, 356, 374, 506-507, 527 in the revised manuscript), as previously defined.
Reviewer Comment: Line 82: Please describe the abbreviation RV-PA.
Response: We have defined the abbreviation RV-PA as right ventricular–pulmonary artery on page 2, line 87-88 in the revised version.
Reviewer Comment: Line 115: You had already used the abbreviation RV-PA, but you did not describe it.
Response: Thank you for pointing this out. The abbreviation RV-PA was defined on page 2, line 87-88, at its first appearance in the manuscript.
Reviewer Comment: Lines 137-238: Please describe the abbreviation.
Response: Thank you for your comment. All abbreviations used in this section, including NT-proBNP and RV-PA, have now been defined at first use.
Reviewer Comment: Lines 237-238: Please describe the abbreviation NT-proBNP.
Response: We have defined the abbreviation NT-proBNP (N-terminal fragment of pro-brain natriuretic protein) in the manuscript at its first mention on page 9, line 273-274.
Reviewer Comment: Line 239: Please use the abbreviations you described.
Response: We confirm that the term RV-PA coupling is used consistently in its abbreviated form following its definition.
Reviewer Comment: (Lines 289, 292, 296, 317, 350, 379): Please use the abbreviations you described.
Response: Thank you. We confirm that the abbreviation PP (pulse pressure) is consistently used as defined (lines 151, 330, 332-334, 336, 363, 380, 412, 452-453, 540 in the revised manuscript).
Reviewer Comment: Line 312 Please correctly discuss the bibliographic citation (17), do not just include a number.
Response: We have revised this sentence as well to appropriately integrate and contextualize the bibliographic reference (17) into the narrative as follows: “This has not yet been described previously and might represent a marker of reduced RV stroke volume and contractility, aligning with the interpretation of proportional PP as a surrogate marker for RV function proposed by Mazimba et al. [1].“ (pages 11-12, lines 365-372)
Reviewer Comment: Line 322 Please use the abbreviations you described.
Response: We confirm that the abbreviations HF (heart failure) and PH (pulmonary hypertension) are used consistently as previously defined.
Reviewer Comment: Line 322 You had already described the abbreviation PH.
Response: Thank you. We confirm that the abbreviation PH continues to be used appropriately following its initial definition.
Reviewer Comment: Line 352 Please correctly discuss the bibliographic citation (17), do not just include a number.
Response: We have revised this sentence as well to appropriately integrate and contextualize the bibliographic reference (17) into the narrative as follows: “This has not yet been described previously and might represent a marker of reduced RV stroke volume and contractility, aligning with the interpretation of proportional PP as a surrogate marker for RV function proposed by Mazimba et al. [1].“ (pages 11-12, lines 365-372). See also comment above.
References:
[1] S. Mazimba et al., “Pulmonary Artery Proportional Pulse Pressure (PAPP) Index Identifies Patients With Improved Survival From the CardioMEMS Implantable Pulmonary Artery Pressure Monitor.,” Heart Lung Circ, vol. 30, no. 9, pp. 1389–1396, Sep. 2021, doi: 10.1016/j.hlc.2021.03.004.
Reviewer 2 Report
Comments and Suggestions for Authors PAP telemonitoring effectively predicts left and global heart failure decompensation, but fails to detect predominantly right-sided congestion. The title does not accurately reflect the content of the study and is initially misleading Patients with right-sided heart failure decompensation were not receiving ARNI therapy — why might that be? The finding on non-HF related hospitalizations is also interesting. Nice use of the torasemide dose equivalent; you might consider also including the furosemide dose equivalent, which can be obtained by simply multiplying by 4. I don’t see any anthropometric parameters; the section on right heart catheterization could be enriched by including PAPI and RVSWI. Have you considered utilizing software tools that enhance the reproducibility of TAPSE and PASP measurements? Recent studies suggest that AI-based software can automate these measurements from echocardiographic images, potentially reducing inter- and intra-observer variability Have you considered including the GLIDE score in the table? It is elegant to express BNP on a logarithmic scale. Excellent statistical analysis, although it remains primarily descriptive and comparative. Have you considered developing a predictive model? Overall, this is an excellent and highly interesting paper that highlights the limitations of implantable invasive monitoring. Could you provide more detail on non-HF-related hospitalizations? Might a drug like vericiguat be useful in preventing right-sided heart failure? Also, data on SGLT2 inhibitors is missing. Great work nonetheless!
Author Response
Reviewer #2:
PAP telemonitoring effectively predicts left and global heart failure decompensation, but fails to detect predominantly right-sided congestion. The title does not accurately reflect the content of the study and is initially misleading Patients with right-sided heart failure decompensation were not receiving ARNI therapy — why might that be? The finding on non-HF related hospitalizations is also interesting. Nice use of the torasemide dose equivalent; you might consider also including the furosemide dose equivalent, which can be obtained by simply multiplying by 4. I don’t see any anthropometric parameters; the section on right heart catheterization could be enriched by including PAPI and RVSWI. Have you considered utilizing software tools that enhance the reproducibility of TAPSE and PASP measurements? Recent studies suggest that AI-based software can automate these measurements from echocardiographic images, potentially reducing inter- and intra-observer variability Have you considered including the GLIDE score in the table? It is elegant to express BNP on a logarithmic scale. Excellent statistical analysis, although it remains primarily descriptive and comparative. Have you considered developing a predictive model? Overall, this is an excellent and highly interesting paper that highlights the limitations of implantable invasive monitoring. Could you provide more detail on non-HF-related hospitalizations? Might a drug like vericiguat be useful in preventing right-sided heart failure? Also, data on SGLT2 inhibitors is missing. Great work nonetheless!
Reviewer Comment: The title does not accurately reflect the content of the study and is initially misleading
Response: Thank you very much for this insightful comment. We have revised the title to more accurately reflect the study's scope and findings. The new title is: “A word of caution – potential limitations of pulmonary artery pressure monitoring in detecting congestion caused by right-sided heart failure.” The term “predominantly” was removed to clearly indicate that episodes of isolated right-sided decompensation might not be reliably detected using this method.
Reviewer Comment: Patients with right-sided heart failure decompensation were not receiving ARNI therapy — why might that be?
Response: We appreciate your observation. The present cohort started inclusion in 2015 and was observed until 2019, during which the adoption of ARNI therapy was still emerging in clinical practice. Entresto® was approved by EMA on 24th November 2015, and was introduced and available in Germany in Q2 2016. In our dataset, at baseline, only 8 patients (21%) were receiving ARNI, whereas at the end of follow-up, this proportion increased to 20 patients (53 %). In 4 cases, the ARNI has been stopped temporarily for hypotension at the end of the observation period.
However, only one patient experiencing a global decompensation event received ARNI therapy already at baseline, prior to decompensation. No patient with isolated left- or right-sided decompensation events were treated with an ARNI at baseline. This likely reflects the limited real-world penetration of ARNIs early after their introduction. In 6 of 12 patients where ARNI has been initiated newly, hospitalizations for cardiac decompensation occurred.
Reviewer Comment: You might consider also including the furosemide dose equivalent, which can be obtained by simply multiplying by 4.
Response: Thank you for this helpful suggestion. We have incorporated the furosemide dose equivalents in Table 1 by applying a conversion factor of 4, as recommended.
Reviewer Comment: I don’t see any anthropometric parameters.
Response: We agree that anthropometric parameters add valuable context to the study population. So that, we added the body mass index in Table 1. At this time, we are unable to perform formal statistical tests to assess significance levels between the groups. Nevertheless, to provide additional insight, we have reported the mean values together with their standard errors for the relevant variables. This allows for a descriptive comparison of the groups.
Reviewer Comment: The section on right heart catheterization could be enriched by including PAPI and RVSWI.
Response: Thank you for this valuable suggestion. We have provided the mean values and standard errors for right ventricular stroke work index (RVSWI) across the respective subgroups. Patients with hospitalizations without cardiac decompensation had an RVSWI of 11 ±â€¯1 g/m². In contrast, patients with cardiac decompensation exhibited an RVSWI of 15 ±â€¯1 g/m². Subgroup analysis showed RVSWI values of 16 ±â€¯2 g/m² in patients with left-sided decompensation, 16 ±â€¯4 g/m² in those with right-sided decompensation, and 15 ±â€¯1 g/m² in cases of global cardiac decompensation. As formal significance testing is not currently feasible due to limited access to statistical support, we have chosen not to include these descriptive RVSWI values in the main manuscript. However, we are providing them here in response to your helpful comment to allow for a preliminary interpretation. Unfortunately, our dataset did not include right atrial pressure, which is required to calculate the pulmonary artery pulsatility index (PAPI). We acknowledge this as a limitation and will ensure that this parameter is collected in future studies.
Reviewer Comment: Recent studies suggest that AI-based software can automate these measurements from echocardiographic images, potentially reducing inter- and intra-observer variability. Have you considered including the GLIDE score in the table?
Response: This is a highly relevant and forward-looking point. While we are aware of the recent advances in AI-assisted echocardiographic assessment, the GLIDE score (which was shown to predict T-TEER success) was not incorporated in this analysis as it was not routinely available in our imaging protocols at the time of data collection. We did not obtain TOE data but only TTE, where the parameters from GLIDE score are less well established. We agree that the inclusion of the GLIDE score might contribute interesting novel insights into the prediction of right sided cardiac decompensation due to tricuspid regurgitation. However, due to availability of data, this has to be analysed in future prospective studies.
Reviewer Comment: Have you considered developing a predictive model?
Response: We appreciate the suggestion. While the primary aim of the current study was to explore the limitations of pulmonary artery pressure monitoring in detecting right-sided decompensation, the development of a predictive model is an excellent idea for future research, particularly as more comprehensive datasets become available. However, our cohort is obviously too small for such an analysis.
Reviewer Comment: Could you provide more detail on non-HF-related hospitalizations?
Response: Thank you for this important comment. The causes of non-heart failure-related hospitalizations in our cohort were heterogeneous and included both elective and acute admissions. Examples include elective procedures such as cardiac resynchronization therapy (CRT) device implantation or change, interventional valve repair, catheter ablation, and orthopedic surgery (e.g., hip replacement), as well as acute conditions such as gastrointestinal bleeding, acute kidney injury secondary to urinary tract infection or dehydration, and episodes of uncontrolled hyperglycemia.
Reviewer Comment: Might a drug like vericiguat be useful in preventing right-sided heart failure?
Response: This is a very interesting point. Vericiguat, a soluble guanylate cyclase stimulator, has demonstrated benefit in patients with worsening chronic heart failure, predominantly of left-sided origin. None of our patients received vericiguat, as the VICTORIA trial was not available at that point. However, meanwhile there are interesting data for example by T. Hashimoto et al., showing that Vericiguat improved RV-PA coupling and induced LV reverse remodelling [1]. Interestingly, these treatment effects were independent of the quadruple therapy and worsening HF events (limitation: retrospective analysis of 63 patients).
In more current studies like the recruiting PASSPORT trial, the effects of clinically indicated vericiguat treatment of RV-PA coupling in patients supported by PAP-guided HF management can be assessed.
Reviewer Comment: Also, data on SGLT2 inhibitors is missing.
Response: You are correct, and we acknowledge this limitation. During the observation period of our study (2015–2019), SGLT-2 inhibitors were only recommended for the treatment of diabetes, and their beneficial effects in heart failure—independent of left ventricular ejection fraction—had not yet been established. Consequently, SGLT-2 inhibitors were not prescribed for heart failure at that time and were therefore not widely represented in our cohort. In our cohort 3 patients with hospitalizations without cardiac decompensation event were treated with a SGLT-2 inhibitor at baseline. Until the end of the observation period, a SGLT-2 inhibitor was initiated in 2 patients with hospitalizations without cardiac decompensation event. Among patients who experienced hospitalization for cardiac decompensation, none received SGLT-2 inhibitors either at baseline or during follow-up. Future studies incorporating more recent patient data will be better suited to assess the effects of SGLT-2 inhibitors on both left- and right-sided heart failure. Emerging evidence suggests that SGLT-2 inhibition may be associated with reductions in pulmonary artery pressure as well as improvements in TAPSE and FAC [2].
References:
[1] T. Hashimoto et al., “Effectiveness of Vericiguat on right ventricle to pulmonary artery uncoupling associated with heart failure with reduced ejection fraction,” Int J Cardiol, vol. 415, Nov. 2024, doi: 10.1016/J.IJCARD.2024.132441.
[2] T. Cinar et al., “Effects of SGLT2 inhibitors on right ventricular function in heart failure patients: Updated meta-analysis of the current literature,” Polish Heart Journal (Kardiologia Polska), vol. 82, no. 4, pp. 416–422, 2024, doi: 10.33963/V.PHJ.100199.
Reviewer 3 Report
Comments and Suggestions for Authors
- Overall, the authors have carefully collected the data. However, the number of records is relatively low considering the scope of this study.
- The study is scientifically sound.
- The conclusion is supported by the analysis results.
- The authors should consider utilizing additional (advanced) analyses to validate the objectives or assumptions of this study.
- Accordingly, the visualization of data and analysis results will be revised and improved.
- The presentation can be revised and improved.
Author Response
Reviewer #3:
Reviewer Comment: Overall, the authors have carefully collected the data. However, the number of records is relatively low considering the scope of this study.
Response: We thank the reviewer for recognizing the thoroughness of our data collection. We acknowledge that the sample size is limited, which is a constraint inherent to the specific inclusion criteria and the highly selected nature of the patient population monitored with implantable pulmonary artery pressure sensors. Nonetheless, we believe that the insights gained from this cohort remain valuable, particularly in light of the paucity of data regarding right-sided heart failure detection through this modality.
Reviewer Comment: The study is scientifically sound.
Response: We appreciate the reviewer’s positive assessment regarding the scientific validity of our study.
Reviewer Comment: The conclusion is supported by the analysis results.
Response: Thank you for this encouraging feedback. We have ensured that the conclusions remain firmly grounded in the presented data and analysis.
Reviewer Comment: The authors should consider utilizing additional (advanced) analyses to validate the objectives or assumptions of this study.
Response: We thank the reviewer for this valuable suggestion. While the primary focus of this study was descriptive and hypothesis-generating in nature, we agree that more advanced analytical methods such as predictive modeling and multivariate analyses in an expanded dataset could further strengthen the findings. However, this is beyond the scope of the present limited data set.
Reviewer Comment: Accordingly, the visualization of data and analysis results will be revised and improved.
Response: Thank you for this valuable suggestion. In response, we have revised and enhanced the data visualizations to improve both clarity and interpretability. Specifically, Figure 3A has been updated and is now presented in high-resolution TIFF format to ensure optimal visual quality. These revisions are reflected in the updated manuscript.
Reviewer Comment: The presentation can be revised and improved.
Response: We have carefully revised the manuscript for improved clarity, logical flow, and conciseness. Particular attention was paid to the Introduction and Discussion sections to better frame the rationale, highlight the novelty, and articulate the implications of our findings.
Reviewer 4 Report
Comments and Suggestions for Authors
This article explores how well pulmonary artery pressure (PAP) monitoring works for detecting different types of heart failure. It shows that PAP monitoring is useful for spotting early signs of left-sided and global heart failure, but not very effective for right-sided heart failure, where PAP levels often don't change much before a patient gets worse. This is an important finding, since it means some patients may not get the early help they need. The study is detailed and uses solid methods, but it's limited by its small size and the fact that it's a retrospective analysis. Also, because doctors adjusted treatment based on PAP readings, some hospitalizations may have been prevented, which affects the results. Overall, the study raises awareness of a gap in current heart failure monitoring and suggests that measuring right heart function more directly could improve patient care.
To improve this manuscript, the authors could simplify the abstract by clearly highlighting the main findings and why they matter for patient care. The criteria used to classify heart failure types (left-, right-, or global) should be explained more clearly, and it would help to show that the classifications were consistent. The discussion could better explain what the results mean for doctors—for example, when extra tests might be needed alongside pressure monitoring. Since treatment changes may have affected the results, this should be more clearly discussed or analyzed. Some figures are hard to read, so clearer graphs and labels would help. The writing could also be made more concise by cutting repetition and overly technical language. The authors should compare their results more directly with other studies to show what’s new. Finally, the conclusion would be stronger if it suggested specific ideas for future research, like combining pressure monitoring with right heart imaging or other tools.
Author Response
Reviewer #4:
Reviewer Comment: This article explores how well pulmonary artery pressure (PAP) monitoring works for detecting different types of heart failure. It shows that PAP monitoring is useful for spotting early signs of left-sided and global heart failure, but not very effective for right-sided heart failure, where PAP levels often don't change much before a patient gets worse. This is an important finding, since it means some patients may not get the early help they need. The study is detailed and uses solid methods, but it's limited by its small size and the fact that it's a retrospective analysis. Also, because doctors adjusted treatment based on PAP readings, some hospitalizations may have been prevented, which affects the results. Overall, the study raises awareness of a gap in current heart failure monitoring and suggests that measuring right heart function more directly could improve patient care.
Response:
We sincerely thank the reviewer for this thorough and insightful summary of our work. We agree that the findings underscore a clinically important limitation in current heart failure monitoring strategies and support the need for enhanced assessment tools, particularly for right-sided heart failure. We have emphasized this point further in the Discussion and Conclusion sections. We also acknowledge the retrospective nature of the study and the small sample size as limitations and have elaborated on how treatment adjustments based on PAP monitoring could have influenced outcomes in the Limitations section.
Reviewer Comment: To improve this manuscript, the authors could simplify the abstract by clearly highlighting the main findings and why they matter for patient care.
Response:
Thank you for this suggestion. We have revised the abstract to be more concise and patient-centered (page 1, lines 39-42). The main findings are now explicitly stated, and the clinical relevance—especially the limitations of PAP monitoring in detecting right-sided decompensation—is clearly highlighted.
Reviewer Comment: The criteria used to classify heart failure types (left-, right-, or global) should be explained more clearly, and it would help to show that the classifications were consistent.
Response:
We appreciate this important point. We have revised the Methods section to provide a more detailed explanation of the clinical, imaging, and laboratory criteria used to classify heart failure episodes as left-sided, right-sided, or global on page 3, lines 126. The classification was performed by a single experienced clinician, which we now explicitly state as a limitation in the revised manuscript on page 14, lines 512-514.
Reviewer Comment: The discussion could better explain what the results mean for doctors—for example, when extra tests might be needed alongside pressure monitoring.
Response:
Thank you for this valuable suggestion. We have expanded the Discussion section to address the practical implications for clinicians (page 14, lines 483–484). In particular, we now emphasize that supplementary assessments (e.g., echocardiography) may be warranted when PAP readings remain stable despite clinical signs of decompensation (page 12, lines 416–419).
Reviewer Comment: Since treatment changes may have affected the results, this should be more clearly discussed or analyzed.
Response:
We agree that physician-initiated treatment modifications in response to PAP trends could have mitigated the progression to overt decompensation and thereby influenced hospitalization rates. This potential confounder is addressed more in the Limitations section (page 14, lines, 514-524) along with a discussion of its impact on interpreting the true predictive value of PAP monitoring.
Reviewer Comment: Some figures are hard to read, so clearer graphs and labels would help.
Response:
Thank you for this helpful feedback. Specifically, Figure 3A has been updated and is now presented in high-resolution TIFF format to ensure optimal visual quality. These revisions are reflected in the updated manuscript.
Reviewer Comment: The writing could also be made more concise by cutting repetition and overly technical language.
Response:
We have carefully edited the manuscript for conciseness and clarity. Redundant passages have been removed, e.g., lines 50, 52-53, 57, 59, 62-63, 66, 68, 76-77, 79, 82, 90, 113, 117-118, 125, 185, 204, 221, 356, 374, 506-507, 527 in the revised manuscript, we used the abbreviation “HF” for Heart Failure. Overly technical terms have been simplified where appropriate, without compromising scientific rigo e.g., in line 87-88 and 126.
Reviewer Comment: The authors should compare their results more directly with other studies to show what’s new.
Response:
We have now strengthened the Discussion by incorporating more direct comparisons with recent published studies. This highlights the novelty of our focus on isolated right-sided failure and the under-recognized challenges it presents (page 12, 395-403).
Reviewer Comment: Finally, the conclusion would be stronger if it suggested specific ideas for future research, like combining pressure monitoring with right heart imaging or other tools.
Response:
We fully agree. The Conclusion has been revised to include specific recommendations for future research directions, including the integration of PAP monitoring with non-invasive assessments of right heart function, such as echocardiographic parameters (e.g., TAPSE, RV-FAC, RV-PA coupling). We also propose the potential role of machine learning approaches to better differentiate patterns of decompensation (page 15, lines 543-544).
Round 2
Reviewer 1 Report
Comments and Suggestions for Authors
Dear authors of the paper "Biomedicinas-3635792," I extend my warm congratulations for the improvements made to your work.
Reviewer 3 Report
Comments and Suggestions for Authors
The manuscript has been sufficiently revised. It is now in an acceptable form.